# Prediction of the Malignancy of a Breast Lesion Detected on Breast Ultrasound: Radiomics Applied to Clinical Practice

**DOI:** 10.3390/cancers15030964

**Published:** 2023-02-02

**Authors:** Luca Nicosia, Filippo Pesapane, Anna Carla Bozzini, Antuono Latronico, Anna Rotili, Federica Ferrari, Giulia Signorelli, Sara Raimondi, Silvano Vignati, Aurora Gaeta, Federica Bellerba, Daniela Origgi, Paolo De Marco, Giuseppe Castiglione Minischetti, Claudia Sangalli, Marta Montesano, Simone Palma, Enrico Cassano

**Affiliations:** 1Breast Imaging Division, Radiology Department, IEO European Institute of Oncology IRCCS, 20141 Milan, Italy; 2Molecular and Pharmaco-Epidemiology Unit, Department of Experimental Oncology, IEO IRCCS, 20141 Milan, Italy; 3Medical Physics Unit, IEO European Institute of Oncology IRCCS, via Ripamonti 435, 20141 Milan, Italy; 4School of Medical Physics, University of Milan, via Celoria 16, 20133 Milan, Italy; 5Data Management, European Institute of Oncology IRCCS, 20141 Milan, Italy; 6Department of Radiological and Hematological Sciences, Catholic University of the Sacred Heart, Largo Francesco Vito 1, 00168 Rome, Italy

**Keywords:** radiomics, breast cancer, breast ultrasound, breast computer aided diagnosis system

## Abstract

**Simple Summary:**

Breast cancer is the most frequent cancer among women: early diagnosis and management of breast lesions are crucial to achieve a better prognosis for patients with this diagnosis. Breast ultrasound (US) is one of the main techniques for the management of breast lesions and it is important in doubtful findings on mammography and in the evaluation of dense breasts. Unfortunately, US has a high rate of false positive and has high operator dependence. Ultrasound CAD (computer-aided diagnosis) and radiomics are newly developed tools that can help solve these issues: this study aims to create a radiomics score from breast US to predict malignancy of a breast lesion, and to also combine this score with CAD and sonographer performances. Finally, we would like to create a prediction tool of US radiomics features combined with CAD, clinical parameters, and Breast Imaging Reporting and Data System evaluation for the prediction of malignancy of breast lesions.

**Abstract:**

The study aimed to evaluate the performance of radiomics features and one ultrasound CAD (computer-aided diagnosis) in the prediction of the malignancy of a breast lesion detected with ultrasound and to develop a nomogram incorporating radiomic score and available information on CAD performance, conventional Breast Imaging Reporting and Data System evaluation (BI-RADS), and clinical information. Data on 365 breast lesions referred for breast US with subsequent histologic analysis between January 2020 and March 2022 were retrospectively collected. Patients were randomly divided into a training group (*n* = 255) and a validation test group (*n* = 110). A radiomics score was generated from the US image. The CAD was performed in a subgroup of 209 cases. The radiomics score included seven radiomics features selected with the LASSO logistic regression model. The multivariable logistic model incorporating CAD performance, BI-RADS evaluation, clinical information, and radiomic score as covariates showed promising results in the prediction of the malignancy of breast lesions: Area under the receiver operating characteristic curve, [AUC]: 0.914; 95% Confidence Interval, [CI]: 0.876–0.951. A nomogram was developed based on these results for possible future applications in clinical practice.

## 1. Introduction

Breast cancer represents the fifth leading cause of death in the global population, with an estimated 2.3 million cases in 2020, and has recently become the most diagnosed cancer, surpassing lung cancer. It is currently the most frequent cancer among women, with a prevalence of 24.5%, and it is the leading cause of mortality (15.5%) [1]. At least one in eight women receive a breast cancer diagnosis in their lifetime [2].

Considering these critical epidemiological data, an early diagnosis and proper management of patients with breast lesions appear to be of huge importance. Full-field digital mammography (FFDM) is the main breast screening method; however, it is known to have low sensitivity in dense breasts. In those patients, the complementary role of breast ultrasound can be of great help [3].

Breast ultrasound (US) is one of the leading traditional techniques used for managing breast lesions and can be used to define doubtful findings of mammography and MRI better and to manage dense breasts better. Indeed, it has been demonstrated that breast US could identify additional breast lesions in dense breasts. According to some recent studies, breast US [4,5] can identify additional cancers in a range from two to seven every 1000 negative mammographs.

One of the main challenges of a breast diagnostic technique is to try to maintain a proper balance between false-positive and false-negative results. Unfortunately, fewer than one of ten biopsies prompted by the US turns out to be malignant [6], with unfavorable consequences for health care costs, waiting list organization, and patient psychological distress.

The high operator dependence of the US shows that the attribution of the risk of the malignancy of a breast lesion is highly subjective and dependent on the operator’s experience [7]. To solve this problem, ultrasound CAD (computer-aided diagnosis) has been developed recently and has achieved good results in clinical practice, although still improvable [7]. An ultrasound CAD uses a deep-learning algorithm to evaluate ultrasound breast lesions. These algorithms analyze the breast lesion from a morphological point of view and indicate the benignity or malignancy of the lesion itself. The CAD we used in this study was the Ultrasonic S-Detect (Samsung Medison Co., Ltd., Seoul, Republic of Korea) [7]. Radiomics can also provide an essential aid in helping to predict the malignancy of breast lesions: the recent strong development of radiomics in clinical imaging, especially in the field of oncology, has presented auspicious results [8,9]: radiomics is a quantitative approach to medical imaging that analyzes the grey values of a radiological image with the extraction, through a computer algorithm, of quantitative information that is not obtainable from conventional qualitative analysis [10,11,12]. Features extracted can be associated with clinically significant outcomes such as the prediction of malignancy of a breast lesion.

Radiomics, through the aid of artificial intelligence, specifically machine and deep learning, enables the computational analysis of medical images, with quantitative features being obtained. These features make it possible to build models that can help improve diagnosis and treatment in oncology.

Moreover, radiomics can have a key development by combining it with clinical practice. Indeed, medical imaging information is related to malignancy structure and behavior [13]. Thus, data obtained from the radiomic process can be analyzed and correlated with clinical events [14,15].

This study aimed to create a radiomics score from the breast US to predict the malignancy of a breast lesion. We also wanted to evaluate whether the performance of this model could be combined with the ultrasound CAD (S-detect) information to improve the overall results obtained in predicting the malignancy of a lesion. We finally aimed to create a tool (nomogram) of ultrasound radiomics features combined with CAD, clinical information, and US Breast Imaging Reporting And Data System (BI-RADS) [13] evaluations to predict the malignancy of the breast lesions in clinical practice.

## 2. Materials and Methods

This retrospective study was submitted to the Ethics Committee (Identification Number UID 2793, 27 July 2021) and approved by the Institutional Review Board.

We reviewed a series from an internal database of patients who underwent ultrasound core needle breast biopsies for suspicious breast lesions (BI-RADS > 3) between 1 January 2020 and 31 March 2022 of 365 images (corresponding to 365 breast lesions from 362 consecutive patients).

All of the pre-biopsy images of the lesions were acquired with a Samsung RSV80A or RSV85 Healthcare, Seoul, Republic of Korea. All 365 selected images were used for radiomic analysis. Manual segmentations were performed with the LifeX program [16] by three different radiologists, as shown in Figure 1.

The images were reviewed according to conventional ultrasound BI-RADS [17] by four different radiologists, two with 5 years of experience in breast imaging and two with more than 15 years of experience.

According to our clinical practice, BI-RADS < 4a was considered predictive of benign lesions, and BI-RADS 4b, c, and 5 were considered predictive of malignant lesions [7,17].

S-detect CAD information (Computer Aided Diagnosis of the Samsung RSV80 or 85 machine, Samsung Medison Co., Ltd., Seoul, Republic of Korea) was available for a subgroup of patients (209). S-Detect (a CAD based on a deep-learning algorithm to evaluate ultrasound breast lesions) was used to obtain a categorization of the breast lesion as either “possibly benign” or “possibly malignant.” For each lesion, the radiologist, after the acquisition of both longitudinal and transverse B mode images, placed a mark in the center of the lesion: the system automatically provided a region-of-interest (ROI) around the border of the mass with a “possibly benign or malignant” score, as shown in Figure 2.

The histopathological biopsy result was used as the gold standard [18] for defining diagnostic accuracy.

### Statistical Analysis

Demographic and clinical characteristics were presented with descriptive statistics, expressed as frequencies and percentages for categorical variables, and as medians and interquartile ranges for continuous variables. Differences among patients by histopathological characteristics were evaluated with Fisher’s exact test for categorical variables or the Wilcoxon rank sum test for continuous variables. The least Absolute Shrinkage and Selection Operator (LASSO) logistic regression model was implemented to select the radiomic features associated with the malignancy of the tumor. The best lambda parameter was estimated using 10-fold cross-validation, while the regularization strength was selected as the minimum value that maximized the area under the curve (AUC).

The radiomic score of each patient was the linear combination of the selected features weighted by their respective LASSO coefficients. Breast lesions were randomly assigned to training and test sets to build the models with a 70:30 rate (255 lesions in the training dataset and 110 lesions in the test dataset). Univariable and multivariable logistic regression models for predicting the malignancy of the lesion (malignant vs. benign) were fitted on the training set and evaluated on the test set. The association of each selected radiomic feature by LASSO with the prediction of malignancy was also evaluated with the Wilcoxon rank sum test for descriptive purposes. The following models were first implemented:-The radiomic model including the radiomic score (calculated with LASSO logistic regression, as described above) as a single covariate;-The Adjusted Radiomic model including the radiomic score and the clinical variables as covariates. We included here the clinical variables associated with malignancy prediction in univariate logistic regression analysis.

Inter-rater agreement among the four radiologists for the BI-RADS score was assessed both overall [Fleiss’s K] [19] and for pairwise operators (weighted Cohen’s K). Moreover, each radiologist’s agreement of the BI-RADS score with the gold standard (histopathology result) was evaluated. The BI-RADS score with the highest agreement with the gold standard was defined as the “best” BI-RADS and was retained in the following analysis incorporating the BI-RADS score:-The Adjusted Radiomic + BI-RADS best model including the radiomic score, the best BI-RADS, and the clinical variables as covariates. For the subgroup of patients with information on S-detect, we also calculated:-The Adjusted Radiomic + S-Detect model including the radiomic score, the S-detect score, and the clinical variables as covariates;-The Adjusted Radiomic + S-Detect + BI-Rads best model including the radiomic score, the S-detect score, the best BI-RADS, and the clinical variables as covariates.

All of the models including the S-detect score as a covariate were estimated, this included only the subgroup of 209 patients for which the S-detect score was available.

Due to the sample size constraint, these latter models were built as subgroup analyses without splitting into testing and training sets.

The predictive performance of the models was evaluated and compared using the area under the receiver operating characteristics curve (AUC); AUC 95% confidence limits were calculated with ‘OptimalCutpoints’ R-package v. 1.1-5. Sensitivity (SE) and specificity (SP) were also calculated using a cut-off that minimized the distance between the ROC plot and the vertex (0,1). To validate the model’s performance in the training cohort, we considered the calibration and discrimination metrics. The overall performance was measured by the Hosmer–Lemeshow test; calibration was assessed considering the intercept and slope calibration. All of the analyses were performed using R 4.1.2 software, and *p* values < 0.05 were considered statistically significant.

## 3. Results

The mean age of the patients was 50 years, and the mean size of the breast masses was 16 mm. A total of 255 patients belonged to the training group, and 110 belonged to the test group; 53% of the lesions were malignant (192 cases): specifically, 173 (47%) were invasive neoplasms, eight (2.2%) in situ lesions, 11 (3%) B3 (uncertain malignant potential) lesions; 173 (47%) were benign findings (Table 1). Histopathological features and outcomes according to the training/test group are shown in Table 1. No significant differences between the histological outcome, age, and lesion size in the training and test groups were found.

After the radiomic evaluation of the US images, the most relevant features were selected in the training dataset. The univariate association of each selected radiomic feature by LASSO with the prediction of malignancy was presented in Table 2: all of the features except for CONVENTIONAL_std showed a significant univariate association with the outcome.

The performances of the radiomic models in the training and test sets are summarized in Table 3. The AUC of the crude radiomic model in the training set was 0.773 (95% CI: 0.716–0.831), the SE was 0.705 (95% CI: 0.619–0.782), and the SP was 0.754 (95% CI: 0.669–0.826). The AUC of the crude radiomic model in the test set was reduced to 0.640 (95% CI: 0.535–0.744), with SE (95% CI): 0.660 (0.517–0.785) and SP (95% CI): 0.614 (0.476–0.740). The AUC of the training-adjusted radiomic model (including age and lesion size as covariates) was increased to 0.842 (95%CI: 0.792–0.891), with SE (95% CI): 0.775 (0.693–0.844) and SP (95% CI): 0.786 (0.704–0.854). The AUC of the test-adjusted radiomic model was 0.781 (95% CI: 0.696–0.865), with SE (95% CI): 0.736 (0.597–0.847) and SP (95% CI): 0.719 (0.585–0.830). Overall, the test and training models showed statistically comparable discrimination performances. The evaluation of good fitting of the radiomic model with the Hosmer test and training group is shown in Appendix A.

The strength of the agreement between the radiologists in the Bi-RADS evaluation was shown in Appendix A. We found an overall good agreement between the four radiologists (Fleiss’s K 0.74, with K = 1 indicating perfect agreement) and excellent agreement among radiologists of similar experience (0.879 for the most expert and 0.830 for the less expert ones). The agreement with the gold standard was good for most expert radiologists (0.62 and 0.66, respectively, for observers 1 and 2) and fair for less expert radiologists (0.37 for observers 3 and 4, results not shown).

Including the US BI-RADS evaluation increased the performance of the radiomics models. The crude model including the BI- RADS evaluation (best observer performances among the four radiologists) showed an AUC = 0.816 (95% CI: 0.769–0.864), the adjusted radiomic model with BI-RADS evaluation (best observer) showed an AUC = 0.918 (95% CI: 0.886–0.951). CAD S-Detect also increased the predictive performance of the radiomic model, although this analysis included only the subgroup of 209 images with scores available (Table 4). Specifically, the crude radiomic model with S-Detect showed an AUC of 0.863 (95% CI: 0.811–0.914), while the adjusted radiomics model with S-Detect showed an AUC of 0.887 (95% CI: 0.840–0.933). For this subgroup of patients, the adjusted radiomic model with the inclusion of BI-RADS assessment (best reader) reached an AUC of 0.883 (0.839–0.927), slightly lower than the AUC reached in the whole sample. Finally, a single model including all the predictors shown thus far (clinical information, radiomics, BI-RADS and S-Detect) was implemented to assess whether the integration of all of these data performed better than the other sub-models. As shown in Table 4, this final model obtained an AUC of 0.914 (95% CI: 0.876–0.951), which was the highest AUC reached in this subgroup of patients. Sensitivity and specificity were also higher than those observed in the other models.

Multivariate logistic regression analyses were used to build a nomogram (Figure 3 for predicting breast malignancy using data from this latter complete model (adjusted radiomic model, S-Detect, and BI-RADS evaluation). In a nomogram, every predictor assesses the total risk based on their specific value graphically reported on a “scale point”. The sum of the values of all the “scales point” imports a “ Total point” value that graphically corresponds to a final “Risk of Outcome”. For example, a patient with a radiomic score value of 0.5 (approx. 40 points), is 65 years old (approx. 50 points), lesion size 45 mm (approx. 10 points), S-Detect and BI-RADS = Benignant (0 points each) collect total points = 100, which corresponds to approx. 15% risk of malignancy.

## 4. Discussion

The considerable development of diagnostic imaging, which we have witnessed in recent years, aims to increasingly improve the performance of diagnostic examinations, making them more objective and less operator dependent. Specifically, in breast imaging, the goal is to increase lesion detection and identify smaller lesions, but also to increase specificity and avoid unnecessary biopsies for benign lesions [20,21]. Ultrasound is one of the fundamental methods for diagnosing breast malignancies, especially in dense breasts where mammography loses sensitivity [22]. Compared with other breast imaging techniques, it is simple to perform, has no radiation, and allows for a real-time evaluation of breast lesions. However, ultrasound has shortcomings such as being extraordinarily operator-dependent and having a narrow field of view [23]. For this reason, it is difficult for radiologists to analyze an ultrasound image thoroughly and reproducibly [5,6]. Efforts have been made in recent years to improve the performance of ultrasound with computerized systems (CAD) [7] and with the application of radiomics to ultrasound images.

With our research, we demonstrated that radiomics applied to ultrasound, using some CAD and known clinical knowledge, can improve performance in predicting the malignancy and benignity of breast lesions.

As in our study, some authors, in a few recently published research papers, have tried to combine radiomics with other US tools that aim to improve ultrasound performance such as shear-wave elastography (SWE) or color-Doppler (CD) [21,24,25,26,27,28,29]. For example, Jiang et al., in a study in 2021 on 401 lesions, showed an AUC in the prediction of malignancy of 0.920, considering the performance of a radiomic model plus shear-wave elastography [30]. Other studies have combined the radiomic model from US B mode images with a different US tool such as color-Doppler, but also with clinical information, in a similar way to our study, like that of Moustafa et al. [31]. They retrospectively analyzed US images of 159 solid masses from 156 patients to build a model including the analysis of B mode and color-Doppler US images with clinical data of age, which gave a very good performance with a calculated AUC of 0.958. Lower but still excellent performance was obtained in other studies in which only the diagnostic performance of the radiomic model was considered [32,33,34,35]. For example, Mango et al., in a study of 900 lesions, reported an AUC of 0.870 [35]. Our study is in line with the results of the few recent studies already published and, to the best of our knowledge, is the first to combine a radiomic model with an ultrasound CAD such as S-Detect. Our average performance of the combined model was also excellent with an average AUC value of 0.914; in Table 5, we compared our study methods and performance, with other similar studies that assessed the use of radiomics in US images for differentiating between benign and malignant lesions.

Following the data obtained, radiomics offers excellent results in the prediction of the malignancy of breast lesions, if combined with clinical variables, with many possibilities for further development in the near future. Finally, we built a nomogram (Figure 3) that can be used in clinical practice for the prediction of breast malignancy using data from the adjusted radiomic model, S-Detect, and BI-RADS evaluation, expressed as these candidate factors: age, largest lesion diameter, radiomics score, S-Detect score, and US BI-RADS evaluation.

Limitations of our study were the retrospective and single-center nature and the relatively limited number of patients involved. Further studies with a prospective enrolment are necessary to confirm our promising results.

## 5. Conclusions

Our radiomic model approach, along with clinically relevant information and the radiological experience of sonographers, showed excellent results in the prediction of the malignancy of breast lesions. Adding the CAD ultrasound system’s performance further increased the prediction accuracy of malignancy.

## Figures and Tables

**Figure 1 cancers-15-00964-f001:**
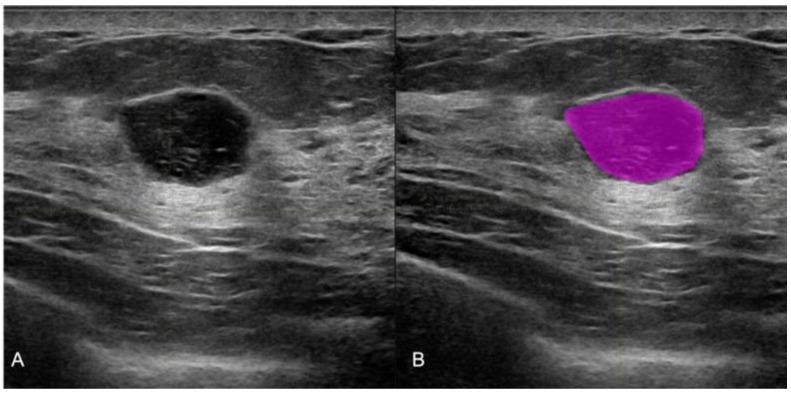
Manual ultrasound image segmentation on LifeX software. (**A**) Round shape breast lesion in the US conventional B mode, (**B**) the manual segmentation with colored ROI obtained with LifeX software. The histological result of the biopsy was that of a fibroadenoma.

**Figure 2 cancers-15-00964-f002:**
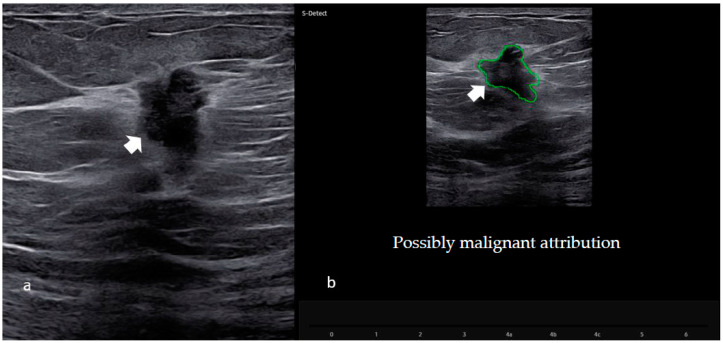
CAD software evaluation (S-detect) in the US images. (**a**) Ultrasound image of the suspicious hypoechogenic nodule with irregular margins in the right breast (as shown by the arrow). (**b**) Region-of-interest (ROI) of the same breast lesion after the S-Detect evaluation. Histological examination of the biopsy confirmed the presence of a malignant neoplasm (invasive ductal carcinoma).

**Figure 3 cancers-15-00964-f003:**
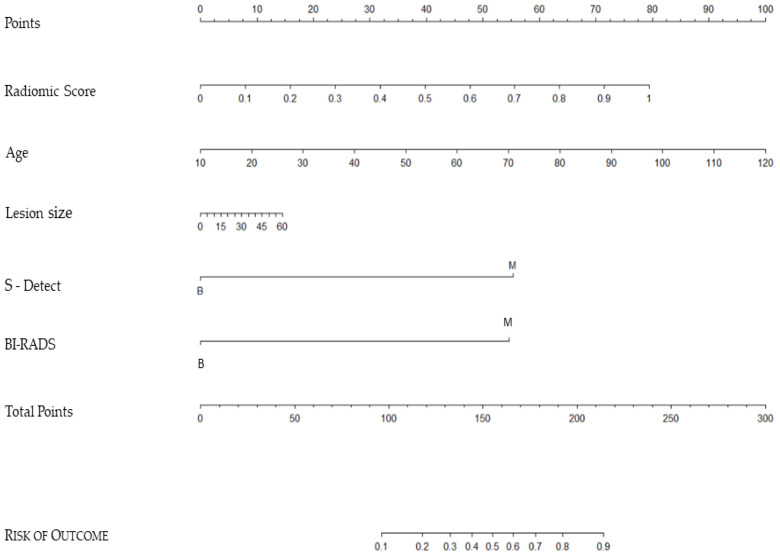
Nomogram for the prediction of breast malignancy using the adjusted radiomic model, S-Detect CAD system, and BI-RADS evaluation.

**Table 1 cancers-15-00964-t001:** Histopathological characteristics in the training and test groups.

Characteristic	Overall, *N* = 365	Test, *N* = 110	Training, *N* = 255	*p*-Value ^2^
Age ^1^ median (IQR)	50 (41–63)	52 (40–65)	50 (42–62)	0.8
Lesion size median (IQR)	16 (12–23)	18 (12–26)	16 (11–20)	0.2
Histopathological category				>0.9
Benign findings (B2) *n* (%)	173 (47%)	54 (49%)	119 (47%)	
Uncertain malignant potential lesions (B3) *n* (%)	11 (3.0%)	4 (4%)	7 (3%)	
In situ neoplasm (B5a) *n* (%)	8 (2.2%)	2 (1.8%)	6 (2.4%)	
Invasive neoplasm (B5b) *n* (%)	173 (47%)	50 (45%)	123 (48%)	

^1^*N* of subjects with data on age: 362. ^2^ Fisher’s exact test for categorical variables; Wilcoxon rank sum test for continuous variables. IQR = interquartile range.

**Table 2 cancers-15-00964-t002:** Univariate association between the selected radiomic features and prediction of malignancy.

Characteristic	Overall, *N* = 255 ^1^	0, *N* = 126 ^1^	1, *N* = 129 ^1^	*p*-Value ^2^
CONVENTIONAL_std	23.4 (20.7, 26.7)	24.1 (20.9, 27.1)	22.9 (20.5, 25.9)	0.10
CONVENTIONAL_Skewness	1.10 (0.74, 1.47)	1.16 (0.84, 1.52)	1.07 (0.64, 1.40)	0.015
CONVENTIONAL_Kurtosis	4.53 (3.47, 5.93)	4.84 (3.75, 6.46)	4.00 (3.23, 5.25)	<0.001
DISCRETIZED_ExcessKurtosis	1.02 (0.26, 2.36)	1.44 (0.49, 2.84)	0.73 (0.02, 2.01)	<0.001
GLCM_Contrast___Variance	6 (4, 12)	8 (5, 13)	5 (3, 10)	<0.001
NGLDM_Busyness	1.53 (0.74, 2.92)	1.30 (0.58, 2.50)	1.83 (0.83, 3.75)	0.019
GLZLM_SZE	0.61 (0.56, 0.67)	0.62 (0.58, 0.68)	0.58 (0.54, 0.66)	<0.001

^1^ Median (IQR). ^2^ Wilcoxon rank sum test.

**Table 3 cancers-15-00964-t003:** Performances of the radiomic models.

Model	AUC (CI 95%)	Sensitivity (CI 95%)	Specificity (CI 95%)
Training—Crude ^1^ Radiomic	0.773 (0.716–0.831)	0.705 (0.619–0.782)	0.754 (0.669–0.826)
Training—Adjusted ^2^ Radiomic	0.842 (0.792–0.891)	0.775 (0.693–0.844)	0.786 (0.704–0.854)
Test Crude ^1^ Radiomic	0.640 (0.535–0.744)	0.660 (0.517–0.785)	0.614 (0.476–0.740)
Test—Adjusted ^2^ Radiomic	0.781 (0.696–0.865)	0.736 (0.597–0.847)	0.719 (0.585–0.830)

^1^ Crude logistic models include only the radiomic score as a covariate; ^2^ Logistic model including the radiomic score, age, and dimension of the lesion as covariates.

**Table 4 cancers-15-00964-t004:** Performance of adjusted radiomic model with S-Detect, with BI-RADS. Evaluation and together with S-Detect and BI-RADS evaluation; dataset of *N* = 209 patients.

Model (Training Group)	AUC	SE	SP
Adjusted Radiomic + S-Detect	0.887 (0.840–0.933)	0.854 (0.771–0.916)	0.802 (0.716–0.873)
Adjusted Radiomic + BI-RADS best	0.883 (0.839–0.927)	0.854 (0.854–0.771)	0.764 (0.672–0.841)
Radiomic + S-Detect + BI-RADS best	0.914 (0.876–0.951)	0.854 (0.771–0.916)	0.849 (0.766–0.911)

**Table 5 cancers-15-00964-t005:** Performance comparison (AUC) between our study and other similar studies on the application of radiomics in US images.

STUDY	Imaging Data and Other Combined Analyzed Data	Data Size	Radiomic Performance (AUC)
Zhang et al., 2019 [28]	B mode US + SWE	227	0.961
Moustafa et al., 2020 [31]	B mode US + CD + clinical data	159	0.958
Jiang et al., 2021 [30]	B mode US + SWE	401	0.920
Romeo et al., 2021 [32]	B mode US	201	0.820
Qian et al., 2021 [29]	B mode US + CD	873	0.922
Qian et al., 2021 [29]	B mode US + CD + SWE	873	0.955
Current study	B mode US + CAD + clinical data	209	0.920

## Data Availability

The data presented in this study are available on request from the corresponding author. The data are not publicly available due to privacy concerns, in accordance with GDPR.

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
