# Peer review of "Prediction of the Malignancy of a Breast Lesion Detected on Breast Ultrasound: Radiomics Applied to Clinical Practice"

_cancers, 2023, doi:10.3390/cancers15030964_

Round 1

Reviewer 1 Report

Minor grammar and English language revision

The main question addressed by this research is if radiomics features and CAD are useful in the prediction of the malignancy of an ultrasound detected breast lesion. 

The topic is relevant in the field, taking into consideration both the incidence and prevalence of breast cancer and the development of artificial intelligence and its usefulness in medical practice.  

The results of the study are consistent with the results of the studies published and the novelty of this study is the combination of a radiomic model with ultrasound CAD. The radiomic score and the nomogram proposed may be of interest for clinical practice, under the same circumstances of analysis.

The methodology used is adequate and the statistical analysis is appropriate. The number of cases studied is statistically significant. Further studies might be considered useful, taking into consideration the fact that scientific evidence is based on large amount of cases studied with consistent results obtained. Standardization and minimizing subjectivity might be also subject for further studies.

The conclusions are consistent with the material and the methodology of the study, stating that the radiomic model proposed, together with the clinical data and the radiologists’ experience can predict the malignancy of an ultrasound detected breast lesion and that the use of a CAD ultrasound system can increase this result.

The references are appropriate and up to date.

The tables synthesize the results of the data studied in an understandable manner for the reader.

Author Response

The main question addressed by this research is if radiomics features and CAD are useful in the prediction of the malignancy of an ultrasound detected breast lesion. 

The topic is relevant in the field, taking into consideration both the incidence and prevalence of breast cancer and the development of artificial intelligence and its usefulness in medical practice.  

The results of the study are consistent with the results of the studies published and the novelty of this study is the combination of a radiomic model with ultrasound CAD. The radiomic score and the nomogram proposed may be of interest for clinical practice, under the same circumstances of analysis.

The methodology used is adequate and the statistical analysis is appropriate. The number of cases studied is statistically significant. Further studies might be considered useful, taking into consideration the fact that scientific evidence is based on large amount of cases studied with consistent results obtained. Standardization and minimizing subjectivity might be also subject for further studies.

The conclusions are consistent with the material and the methodology of the study, stating that the radiomic model proposed, together with the clinical data and the radiologists’ experience can predict the malignancy of an ultrasound detected breast lesion and that the use of a CAD ultrasound system can increase this result.

The references are appropriate and up to date.

The tables synthesize the results of the data studied in an understandable manner for the reader.

We thank the reviewer for comments on the study that gratify our work.

Reviewer 2 Report

Thank you very much for inviting me to review this interesting paper. It is and important and promising issue.

Here are my observations:

In general, you should review the English, there are some minor corrections to be made, including Grammar, commas and periods.

Line 45: AUC you wrote the number with a comma, you should change for a period 0.9714 (you wrote 0,914)

Line 46: in twice

Line 63: it has been demonstrated (take out the "now", it is not news)

Line 74: this problem (not kind)

Line 111: I don't think you should say less or more experience, just say how many years of experience

Line 122: "possibly benign or malignant” evaluation - I think it would sound better score instead of evaluation

Line 128: why micro? We usually say histopathological. Change for "was used", not has been

Line 181: reformulate sentence in better English

Line 183: you should specify what are B3 lesions in the text (lesions of uncertain malignant potential)

How was the training group evaluated?

Table 1 - age and size - I suppose you wrote the range in parentehsis, use a hyphen instead of a comma and explain it in the table, also say if it is median, and inferiorly say this is the n

Discussion: from line 247 to 269' you are talking background of radiomics. You should start your discussion part with a discussion of what you have discovered and then talk generally, but part of this should be in the introduction.

Table 5: AUC - use period not comma in the number

Line 299: single center is better than monocentric

Line 304: prediction of malignancy - take out of the

Line 306: increased

I think you should talk more about your results in the discussion and compare to other studies

Author Response

Thank you very much for inviting me to review this interesting paper. It is and important and promising issue.

Thank you for your comment.

We are also convinced of the interest and relevance of the topic.

Here are my observations:

In general, you should review the English, there are some minor corrections to be made, including Grammar, commas, and periods.

Thank you for the comment.

A native English speaker revised the text.

Line 45: AUC you wrote the number with a comma, you should change for a period 0.9714 (you wrote 0,914)

Thanks, we corrected the text according to your suggestion.

Line 46: in twice

Thanks, we corrected the text and we are sorry for the typo.

Line 63: it has been demonstrated (take out the "now", it is not news)

Thanks for the suggestion, we corrected the text.

Line 74: this problem (not kind)

Thanks, we corrected it according to your suggestion.

Line 111: I don't think you should say less or more experience, just say how many years of experience

Thanks, we corrected the text according to your suggestion.

Line 122: "possibly benign or malignant” evaluation - I think it would sound better score instead of evaluation

Thanks, we corrected the text according to your suggestion.

Line 128: why micro? We usually say histopathological. Change for "was used", not has been

We are sorry for the typo, we corrected it accordingly.

Line 181: reformulate sentence in better English

Thanks for the suggestion. We revise the English of the whole text.

Line 183: you should specify what are B3 lesions in the text (lesions of uncertain malignant potential)

Thanks for the suggestion. We better specified the meaning of B3 lesions in the text.

How was the training group evaluated?

The belonging of the study patients to the training group was randomly decided. We have better highlighted this part in the text

To build the models, breast lesions were randomly assigned to training and test sets with 70:30 rate (255 lesions in the training dataset and 110 lesions in the test dataset)

Table 1 - age and size - I suppose you wrote the range in parentehsis, use a hyphen instead of a comma and explain it in the table, also say if it is median, and inferiorly say this is the n

Thanks, we modified table 1 according to your suggestion. Descriptive statistics were calculated on 365 lesions from 362 patients. The overall number of lesions is in the column heading. We have now added to the note the N for patients.

Discussion: from line 247 to 269' you are talking background of radiomics. You should start your discussion part with a discussion of what you have discovered and then talk generally, but part of this should be in the introduction.

Thank you for the suggestion.

We have changed the discussion according to your suggestions and moved the descriptive part about radiomics to the introduction.

Table 5: AUC - use period not comma in the number

Thanks, we changed the text accordingly.

Line 299: single center is better than monocentric

Thanks, we changed the text accordingly.

Line 304: prediction of malignancy - take out of the

Thanks, we changed the text accordingly.

Line 306: increased

Thanks, we changed the text accordingly.

I think you should talk more about your results in the discussion and compare to other studies

Thank you for your suggestion.

There is a full discussion of how the results of our study fit into the landscape of what has already been published.

Table 5 summarizes this comparison succinctly.